# Enhancing the Dispersion Stability and Sustained Release of S/O/W Emulsions by Encapsulation of CaCO_3_ Droplets in Sodium Caseinate/Xanthan Gum Microparticles

**DOI:** 10.3390/foods11182854

**Published:** 2022-09-15

**Authors:** Jie Zhang, Gongwei Li, Duoxia Xu, Yanping Cao

**Affiliations:** Beijing Advanced Innovation Center for Food Nutrition and Human Health (BTBU), School of Food and Health, Beijing Engineering and Technology Research Center of Food Additives, Beijing Higher Institution Engineering Research Center of Food Additives and Ingredients, Beijing Key Laboratory of Flavor Chemistry, Beijing Laboratory for Food Quality and Safety, Beijing Technology & Business University, Beijing 100048, China

**Keywords:** calcium carbonate, solid/oil/water emulsions, slow- release, in vitro digestion

## Abstract

In this study, solid/oil/water (S/O/W) emulsions were prepared by sodium caseinate (NaCas) and Xanthan gum (XG) binary composite to improve the dispersion stability of calcium carbonate (CaCO_3_) and achieve a targeted slow-release effect. CaCO_3_ S/O/W emulsions were determined by particle size, Zeta potential, physical stability, and microstructure. X-ray diffraction (XRD), Raman spectroscopy, and Fourier transform infrared spectroscopy (FTIR) were used to characterize the molecular interactions among components. Molecular docking technology was used to predict the possible binding mode between NaCas-XG. The percentage of free Ca^2+^ released in the gastrointestinal tract (GIT) model was also studied. It was found that when the concentration of XG was 0.5 wt% and pH was 7, the particle size was smaller, the distribution was uniform, and the physical stability was improved. The microstructure results showed that the embedding effect of S/O/W emulsions was better, the particle size distribution was more uniform when XG concentration increased and formed a filament-like connector with a relatively more stereoscopic structure. XRD results confirmed that the CaCO_3_ was partially covered due to physical embedding. Infrared and Raman analysis and molecular docking results showed electrostatic and hydrophobic interaction between NaCas and XG. In the GIT digestion model, S/O/W emulsion released Ca^2+^ slowly in the gastric digestion stage, which proved the targeted slow-release effect of the S/O/W emulsions delivery vector. The results showed that the S/O/W emulsions delivery system is an effective way to promote the application of CaCO_3_.

## 1. Introduction

Calcium carbonate (CaCO_3_) is the first choice of calcium supplement raw materials because of the highest calcium content (40% *w/w*), low price, and chemical stability [1,2,3]. It is generally considered to maintain the growth and development of the body and play an essential physiological regulation role in the function of all human cells [4]. However, CaCO_3_ has some problems, such as poor suspension stability and easy precipitation, long-term use will consume a large amount of gastric acid and cause indigestion and abdominal distension, and other problems, which seriously limits its application in the food industry [5]. Given the existing problems of CaCO_3_, it is a complex problem to develop a new carrier for the food nutrition delivery system to improve its quality and the efficacy of targeted and sustained release. At present, studies on CaCO_3_ delivery carriers mainly include nano CaCO_3_-sodium alginate/gelatin microspheres [6], nano-CaCO_3_ Pickering emulsions [7], and micro powdered calcium powder [8], which effectively improve storage stability and calcium absorption rate. However, there are few studies on the traditional CaCO_3_ particles in food emulsions to reduce irritation during stomach digestion [9]. The study found that S/O/W three-phase carrier microspheres are a new solid-phase nutrient delivery system with low preparation cost and simple operation. Currently, studies on the protection of nutrients such as enzymes, vitamins, and probiotics are carried out using S/O/W technology [10,11,12]. A study found that konjac gum S/O/W microspheres can make konjac gum slowly released in the gastrointestinal tract to achieve satiety and reduce appetite [13]. The delivery of CaCO_3_ by S/O/W emulsion is a way to improve dispersion stability and achieve targeted controlled release in liquid food.

Food emulsion is a complex dispersion system formed by shearing and homogenization with protein, polysaccharides, and oil as raw materials. In general, polysaccharides and proteins play an essential role in the production and stabilization of emulsions, mainly because they can reduce the interfacial tension of droplets and improve the viscosity of the continuous phase [14]. Xanthan gum (XG) and sodium caseinate (NaCas) are commonly used to enhance the stability and texture of food. XG is an anionic polysaccharide widely used in food products due to its specific physical (viscosity, pseudo-plasticity, suspension, emulsification) and chemical (water solubility, pH stability) properties; it can significantly increase the viscosity of the dispersed phase of the emulsion or form gel network structure to reduce the instability caused by gravity or Brownian motion and give the product ideal texture characteristics [15,16]. Due to its nutritional (rich in various amino acids needed by the human body) and functional (thickening, emulsification) importance, NaCas is used as an ingredient in a wide range of food products [17,18]. Protein and polysaccharides play an essential role in forming the structure of the food system and maintaining the stability of products. They have specific applications in improving quality and extending shelf life. Studies have shown that the colloid concentration affects XG’s stabilizing effect. The interaction of NaCas and polysaccharide interface and its influence on the stability of emulsion indicates that the system’s pH value, salt, and polysaccharide concentration significantly affect the phase separation behavior of the protein-polysaccharide complex system [19].

Therefore, the purpose of this study was to inquire into the ability of NaCas-XG as the W phase to improve the physicochemical stability of S/O/W emulsions and the slow release of free Ca^2+^ in gastric digestion. The particle size, Zeta potential, physical stability, apparent viscosity, and microstructure were determined in the S/O/W emulsions. Raman spectroscopy, Fourier transform infrared spectroscopy (FTIR), and X-ray diffraction (XRD) were used to characterize the molecular interactions among components. Molecular docking technology was used to predict the possible binding mode between NaCas-XG, and the interaction between NaCas and XG was visualized. The percentage of free Ca^2+^ released in the GIT model was also studied. This study has specific guiding significance for understanding the relationship between protein-polysaccharide interaction and emulsion stability and the rational utilization of XG as a food thickener and stabilizer and provides scientific theory and technical method for calcium delivery.

## 2. Materials and Methods

### 2.1. Materials

CaCO_3_ was obtained from Shuang Teng Industrial Co., Ltd. (Henan, China). Sodium caseinate (Lot#C10185129) and Xanthan gum were purchased from Macklin Biochemical Co., Ltd. (Shanghai, China). Soybean Oil was supplied from COFCO Co., Ltd. (Tianjin, China). Nile Blue A and Nile Red were purchased from Sigma-Aldrich Co. (St. Louis, MO, USA). The stomach pepsin (Lot#L22M9J56574, the activity was 1:10,000) from porcine gastric mucosa, lipase (Lot#T17N9C75063, BR, the activity was 30,000 U/g) from porcine pancreas and bile salt (Lot#K01N8B47111, cholic acid content was ≥60%) were purchased from Yuanye Bio-Technology Co., Ltd. (Shanghai, China). All other chemicals were analytical grade purchased from Sinopharm Group Co., Ltd. (Beijing, China).

### 2.2. Preparation of S/O/W Emulsions

W phase preparation: NaCas (2.0 wt%) and XG solutions (0.1, 0.2, 0.3, 0.4, and 0.5 wt%) were prepared separately with PBS buffer (1.0 mM, pH 7.0), stirred in a water bath at 50 °C for more than 2 h, and stored overnight at 4 °C.

Preparation of S/O phase suspension: CaCO_3_:Soybean oil (1:10, *w/w*) was magnetically stirred for 1 h and then stirred again at 15,000 rpm for 3 min using a high-speed mixer (ULTRA TURRAX T25 Digital, IKA, Staufen, Germany) to form S/O phase suspension.

Preparation of S/O/W emulsions: Different concentrations of XG were added to NaCas solution at 1:1, and the mixture was thoroughly mixed under the 50 °C water bath with magnetic stirring for two hours. The 5% S/O phase was then mixed with the 95% W phase (NaCas-XG mixture) and stirred at 15,000 rpm for 5 min. The pH values of S/O/W emulsions were adjusted with 2.00 mol/L HCl and NaOH [8,9,19].

### 2.3. Zeta-Potential, Particle Size, and Physical Stability Measurements

Zeta-potential was measured using Zetasizer Nano-ZS90 (Malvern Instruments, Worcestershire, UK). Before the measurement, the samples were diluted 400 times with 1 mM phosphate buffer (pH 7.0) to reduce the influence of multiple light scattering on measurement errors. Carefully injected the samples into a capillary cell equipped with two electrodes, avoiding bubbles that affect the experimental results. The refractive index ratio was 1.45; the samples were equilibrated for about 120 s. Particle size was measured using a laser diffraction instrument (S3500 Microtrac Inc, Largo, FL, USA) with the refractive index (1.51) and that of the dispersion medium (1.33). Physical stability was tested using the LUMiSizer (LUM GmbH, Berlin, Germany) with the following parameters: the injection volume was 0.4 mL, speed for 2000 rmp, time interval 10 s, and the test temperature was 25 °C. All measurements were prepared at least in triplicate.

### 2.4. Viscosity Tests

The apparent viscosity was performed using a HAAKE rheometer (MARS IQ Air, HAKKE, Dreieich, Germany). Avoid any air bubbles when adding samples to avoid affecting the experimental results. The plate rotor was CC25 DIN, the shear rate was 2–200 S^−1^, and the temperature was maintained at 25 °C during the test.

### 2.5. Microstructure Analysis

#### 2.5.1. Confocal Laser Scanning Microscopy (CLSM)

Confocal scanning laser microscopy (CLSM) (FV3000, Olympus, Japan) was performed on the microstructure of samples. Nile red was used to dye the O phase, and the laser excitation source was 488 nm; Nile blue A was used to dye the W phase, and the laser excitation source was 635 nm. The measurement was observed using a 10× eyepiece and a 60× objective lens.

#### 2.5.2. Cryo-Scanning Electron Microscopy

Cryo-scanning electron microscopy (Cryo-SEM) (Helios NanoLab G3 UC, FEI, Hillsboro, OR, USA) was performed on samples’ cross-sectional and interfacial structures. Emulsion samples were immersed in liquid nitrogen and then transferred to a cryo-preparation chamber (PP3010T, Quorum Technologies, Lewes, UK) under vacuum. The samples were pre-frozen, fractured, and sublimated, then sprayed with gold for microscopic observation. The observation was carried out at a distance between 3 and 5 mm with TLD detection at 2 kV.

### 2.6. X-ray Diffraction (XRD)

The samples were analyzed by an X-ray diffractometer (BRUCKER D8 ADVANCE, Brooke, Germany). The samples were subjected to vacuum freeze-drying (−80 °C, 48 h) to obtain powdered. Test parameters: the spectral range (5–50 degrees), the scanning rate was 2 degrees/min, the acceleration voltage was set to 40 kV, and the tube current was set to 40 mA.

### 2.7. Infrared Spectral

The samples were analyzed by an FT-IR Spectrometer (IS10, Thermo Nicolet Corporation, Waltham, MA, USA). The samples were vacuum freeze-dried (−80 °C, 48 h) to obtain the powder form. Test parameters: the wavenumber range (400–4000 cm^−1^), the resolution was 4 cm^−1^, the signal to emission ratio was set to 50,000:1, and scanned 64 times.

### 2.8. Raman Spectral Analysis

A Raman microspectroscopy system (inVia-Qontor, Renishaw, UK) was used to analyze the samples. The test parameters: were the wavenumber range (100–3300 cm^−1^), the laser light source was 780 nm, the laser power was 18 mW, the exposure time was 10 s, and the exposure was three times.

### 2.9. Molecular Interconnection

Molecular docking techniques were used to elucidate the possible binding modes of receptors (NaCas) and ligands (XG) and receptors (XG) and ligands (NaCas). The 3D structure PDB files of NaCas and XG were searched and collected from the https://pubchem database (accessed on 1 September 2004). Use AutodockTools-1.5.7 software (San Diego, CA, USA) to hydrogenate, charge core and treat non-polar hydrogen. Autodock Vina 1.5.7 was used for molecular docking, and affinity (kcal/mol) was used as an indicator to determine the interaction mode between receptor and ligand.

### 2.10. Simulated Gastrointestinal Tract (GIT) Model

The GIT model was used to study the gastrointestinal digestion of CaCO_3_ S/O/W emulsions involving the oral, gastric, and small intestinal phases [20]. An automated in vitro digestive system (GI20, NI Instruments Australia, Ryde, Australia) was used in this experiment.

Oral stage: initial sample/simulated saliva (1:1) mixed, adding 4 mL SSF and 1 mL deionized water to 5 mL initial sample, the pH was adjusted to 7.0, and the mixture was simulated in vitro for 10 min by continuous shaking (100 rpm) of the digestive system at 37 °C.

Stomach stage: oral digested sample/simulated gastric fluid (1:1) mixed, adding SGF (8 mL) and pepsin (1.6 mL, 3.2 mg/mL) to the oral digestive products. After the pH was adjusted to 3.0, the mixture was shaken for 2 h (100 rpm, 37 °C).

Small intestine stage: stomach digestion samples/intestinal digestive fluid (1:1) mixed, and then SIF (8.0 mL), pancreatic lipase (5.0 mL, 24 mg/mL), bile salts (3.0 mL, 53.6 mg/mL) and deionized water (4 mL) were added to the gastric digestion products. The pH was adjusted to 7.0; the mixture was continuous shaking for 2 h (100 rmp, 37 °C).

#### 2.10.1. Zeta-Potential and Particle Size of the Digested Samples

The experimental method was the same as in Section 2.3.

#### 2.10.2. CLSM of the Digested Samples

The microstructure analysis of digestion products at different stages was the same as in Section 2.5.1.

#### 2.10.3. SDS-PAGE

SDS-PAGE was used to detect the degree of hydrolysis of the W phase at different digestion stages [21]. The glue concentration was 12.5%, the voltage was 110 V, and the sample loading was 20 ug.

#### 2.10.4. Determination of Free Calcium

Free calcium (Ca^2+^) was analyzed using a Model 402 ISE (Jiangfen Electroanalytical Instrument Co., Taizhou, China). A series concentration of Ca^2+^ was prepared, and the electromotive force (EMF) value was measured to obtain the standard curve. The EMF value measured by the sample was brought into the standard curve and converted into the concentration of free Ca^2+^ [22].

### 2.11. Statistical Analysis

Every experiment was performed at least in triplicate, and results were expressed as the mean ± standard deviation of the measured values. Plots and data analysis were performed using Origin 8.5 software and the SPSS 17.0 statistical analysis system (SPSS Inc., Chicago, IL, USA). The obtained results were statistically analyzed using a one-way analysis of variance (ANOVA) with a significance level (P) of 0.05.

## 3. Results and Discussion

### 3.1. Effects of XG Concentrations on CaCO_3_ S/O/W Emulsions

Figure 1a shows the particle size results of S/O/W emulsions with different XG concentrations. The particle size of the S/O/W emulsion was relatively large at a lower XG concentration (0.1–0.2 wt%). With the increase in XG concentration (0.3–0.5 wt%), the particle size decreased significantly, which may be the result of the weak non-covalent complex formed by the combination of NaCas and XG through hydrophobic or electrostatic interaction. However, when the XG concentration was 0.4 and 0.5 wt%, the particle size tended to be stable, and there was no significant difference. The size of the emulsion droplet is closely related to the stability of the emulsion. Stoke’s theory pointed out that if the viscosity and density difference between the system’s dispersed phase and the continuous phase is fixed, the smaller the emulsion particle size, the more stable the system will be [23]. Generally, the greater the absolute value of Zeta potential, the more stable the system [24]. Figure 1b showed that the XG concentration increased, and the absolute value of Zeta-potential of S/O/W emulsions increased slightly but not significantly, indicating that the XG concentration increase did not significantly affect the surface potential of microspheres [19]. The instability index can characterize the stability of emulsions most directly. The better the stability of the emulsion with the lower instability index. Figure 1c shows that the instability index of S/O/W emulsion gradually decreased with the XG concentration. The lowest value (about 0.01) was reached when XG concentration was 0.5 wt%. Because the XG concentration was higher, the viscosity increased, conducive to improving the system’s stability.

### 3.2. Effects of pH on CaCO_3_ S/O/W Emulsions

Figure 2a shows the particle size of the S/O/W emulsions with the pH changes. At pH 5, the particle size of the S/O/W emulsion was larger than 10μm, which may be attributed to the denaturation precipitation of NaCas and the addition of anionic polysaccharide XG to accelerate the aggregation of the emulsion droplets, which was consistent with the results reported previously [19]. In the range of pH 6–9, the pH value of the system has little effect on the particle size of the emulsion, which is about 2.0 μm. Figure 2b shows that the surface potential of the emulsion droplet did not change significantly at pH 7–9. Still, with the further decrease of pH, the electrostatic repulsion between droplets decreased due to the reduction of net negative charge on the protein surface. Under the condition of pH 5, due to the thermodynamic incompatibility between NaCas and XG and the non-directed complexation between the two polymers with opposite charges, sponge-like substances were generated. XG has excellent pH stability, and the stability of the S/O/W emulsion was better in the range of 5–9 [19,25].

### 3.3. Viscosity Analysis

The apparent viscosity is an important index to characterize the rheology of the emulsions and is closely related to the system stability. An effective way to improve the stability of the emulsions is to increase the viscosity. Some studies showed that the higher the viscosity of the emulsion, the higher the XG concentration and the more stability of the system. XG is a typical pseudoplastic fluid whose viscosity decreases with the increase of shear rate. This pseudo-plasticity is very effective in stabilizing suspensions and emulsions [26]. Due to the pH stability of XG, the effect of pH on the viscosity of S/O/W emulsions had not been studied. Here, the impact of different XG concentrations on apparent viscosity with pH 7 was studied, and the result is shown in Figure 3. The XG concentration increased, and the apparent viscosity increased. XG molecules played a thickening role in the system, which improved the stability of the S/O/W emulsion. The higher the shear rate caused the particles to flow in a direction, the flow resistance decreased, and then the apparent viscosity was lower. The viscosity decreased with the shear rate, indicating that the S/O/W emulsion has shear-thinning fluid characteristics. The higher the shear rate, the particles flow in one direction, the flow resistance decreases, and the apparent viscosity decreases. As the shear thinning effect reduced the viscosity of XG, the viscosity of the S/O/W emulsion decreased considerably [19].

### 3.4. Microstructure Analysis

#### 3.4.1. CLSM

CLSM was used to observe the morphological characteristics of S/O/W emulsions from a relatively large field of vision (Figure 4a). Soybean oil presented green, NaCas molecules presented red, and CaCO_3_ molecules (without dye) exhibited black in the CLSM image [27]. The results showed that the XG concentration increased, the particle size decreased, and the distribution was more uniform, consistent with Figure 1. At low XG concentration, the size distribution was wide, the system’s viscosity was low, and the embedding effect was not good. With the increase of XG concentration, the interaction between XG and NaCas increases, the system viscosity is higher, the stability is enhanced, and the embedding effect is improved. The agglomeration phenomenon may even occur when XG concentration is too high. It may be that the high viscosity at the high XG concentration leads to the movement of proteins on the surface of the oil droplet and the obstruction of adsorption [28].

#### 3.4.2. Cryo-SEM

Cryo-SEM showed the cross-sectional structure and adhesion of S/O/W emulsions and observed the microstructure from a microscopic perspective (Figure 4b). XG solution molecules can support solid particles, droplets, and bubble forms due to their ability to form super-bonded ribbon-like spiral copolymers and fragile colloidal network structures, showing strong emulsifying stability and high suspension capacity. When the concentration of XG was low, the overall structure of S/O/W emulsion particles was loose, and the microspheres were connected by incomplete filaments. The S/O/W emulsion was unstable and did not have a good encapsulation effect on CaCO_3_. With the increase of XG concentration, S/O/W emulsions gradually formed dense filaments, and microsphere droplets were evenly distributed in them, which was conducive to improving the stability of S/O/W emulsion. CLSM observed the CaCO_3_ embedding, and Cryo-SEM observed the core-shell structure, proving that S/O/W emulsion microspheres were successfully constructed. Adding CaCO_3_ powder directly into the S/O/W emulsion liquid system provides a theoretical basis for solving the dispersion stability of insoluble calcium salt in liquid food.

### 3.5. XRD Analysis

XRD was used to characterize the embedding situation of CaCO_3_ in S/O/W emulsions, and the results are shown in Figure 5a. The diffraction peaks of CaCO_3_ (S phase): 23, 29, 36, 39, 43, 47.5, and 48.5, respectively [29]. The diffraction peaks of NaCas (W_1_ phase) were 10 and 20, the XG (W_2_ phase) was 20.2, and the NaCas-XG (W phase) were consistent with NaCas. The diffraction peaks of S/O/W emulsions were 20, 23, 29, 36, 39, 43, 47.5, and 48.5, respectively, which were the combination of CaCO_3_ and matrix, and the peak intensity decreased. XRD results showed that the characteristic peak of CaCO_3_ was covered due to physical embedding, indicating that S/O/W emulsions have a certain embedding effect on CaCO_3_.

### 3.6. Infrared Spectral Analysis

Infrared spectral results are shown in Figure 5b. In the infrared spectrum of CaCO_3_ (S phase), the stretching vibration peak of C-O was 1796.99 cm^−1^ (V_4_), and the asymmetric stretching vibration peak of the C-O bond was 1456.26 cm^−1^ (V_3_). The out-of-plane bending vibration peaks of the C-O bond were 872.54 cm^−1^ (V_2_) and 712.67 cm^−1^ (V_1_) [30]. In NaCas (W_1_ phase), O-H bond stretching vibration occurs at 3279.81 cm^−1^, C-H bond stretching vibration occurs at 2958.18 cm^−1^, and amide I band (C=O stretching vibration) occurs at 1600–1700 cm^−1^. The amide II band (N-H deformation vibration, C-H stretching vibration band) appeared at 1400–1550 cm^−1^. XG (W_2_ phase) has two strong characteristic absorption peaks at 1600.54 cm^−1^ and 1405.56 cm^−1^, representing the asymmetric and symmetric stretching vibration of -COO- respectively, and the absorption peak at 1018.71 cm^−1^ represents the stretching vibration of C-O. In the map of the W-phase complex, the positions of some absorption peaks changed, and the stretching vibration absorption peak of complex -OH shifted from 3295.16 cm^−1^ to 3274.73 cm^−1^, indicating hydrogen bond interaction between NaCas and XG [31].

With the addition of soybean oil, the stretching vibration absorption of O/W emulsion -OH was wide in peak-peak deformation and moved towards a high wave number, and its strength decreased, while the stretching vibration of -CH_2_ was wide in peak-peak deformation and moved towards high wave number, and its strength decreased. The peak shape of S/O/W emulsions and W phase was similar, indicating that the S/O phase was located inside the W phase, the W phase was better encapsulated by the S/O phase, and S/O/W emulsions were successfully constructed. This result was consistent with the microstructure result in Figure 4, which was also illustrated by previous studies [8].

### 3.7. Raman Spectrum Analysis

The Raman spectrum results are shown in Figure 5c. The characteristic peak of S phase CaCO_3_ was at 1085 cm^−1^ and 710 cm^−1^ [32]. But when mixed with the O phase, the peak strength weakens, indicating that CaCO_3_ was mostly in oil [8,9].

In the Raman spectrum of NaCas, 850 and 830 cm^−1^ were important bands that can explain the tyrosine residue model, and the characteristic peak of the amide I band was evident at 1662 cm^−1^ [33]. XG showed -OH stretching vibration peak at 3424 cm^−1^, -CH_2_ stretching vibration peak at 2924 cm^−1^, -C=O stretching vibration peak at 1728 cm^−1^, and acetal vibration absorption peak at 1066 and 1024 cm^−1^.

When the S/O phase mixed with the W phase, the peak intensity weakened, and the peak shape was the similarity to the W phase, which indicated that the S/O phase was located inside the W phase, and the W phase was better encapsulated by the S/O phase, and S/O/W emulsions were successfully constructed. This result was consistent with the microstructural results in Figure 4, as indicated by the experimental results of infrared spectroscopy.

### 3.8. Molecular Docking Analysis

Molecular docking technology was used to visualize the interaction between NaCas and XG, and the results are shown in Figure 6. When NaCas was the acceptor and XG was the ligand, the binding energy of NaCas and XG was −1.2 kJ/mol. The hydrogen bond lengths of O_1_ on NaCas and O_1_ and O_2_ on XG were 2.82 Å and 2.90 Å, respectively, indicating strong hydrogen bond interaction between NaCas and XG. When XG was the receptor and NaCas was the ligand, the binding energy of XG and NaCas was −1.2 kJ/mol, and there was hydrophobic interaction. The prediction model of molecular docking was consistent with the results of the Infrared spectrum analysis. The W phase was hydrogen bonding when the NaCas-XG protein-polysaccharide binary complex might contribute to the stability of S/O/W emulsion.

### 3.9. Digestion Characteristics Analysis

#### 3.9.1. Particle Size and Zeta-Potential of Different Digestion Stages

Figure 7 shows the samples’ particle size (Figure 7a) and Zeta-potential (Figure 7b) at different digestion stages.

The average particle size of the samples decreased after oral digestion. Since NaCas and XG are negatively charged, and the intermolecular electrostatic repulsion is sufficient to overcome the van der Waals force attraction and hydrophobic interaction, they cannot interact to form a unique three-dimensional structure. This results in a reduction in the average particle size [8].

After simulated gastric digestion, the average particle size increased, possibly due to decreased electrostatic repulsion and changes in pH and ionic strength leading to protein hydrolysis by pepsin. It was speculated that the stable double helix structure of XG makes it have strong resistance to enzymatic hydrolysis, so it cannot be degraded by pepsin [34,35]. After simulated intestinal digestion, the average particle size was 3.10 ± 0.30 um, indicating that S/O/W emulsion was further digested and decomposed in the small intestine. The particle size at different digestion stages was closely related to the degree of digestion at each stage, which was related to the composition of the W phase. It was consistent with the microstructure results in Figure 8.

After oral digestion (pH 7.0), the absolute value of Zeta-potential decreased to a certain extent because of the dilution when mixing the emulsions with simulated saliva. After gastric digestion (pH 3.0), the Zeta-potential decreased because of the electrostatic shielding effect of the reduced pH and the relatively high ionic strength in simulated gastric juices. After intestinal digestion (pH 7.0), the Zeta- potential remained negative due to increased pH and FFA production, possibly because of the influence of anionic fatty acids and bile salts in intestinal digestive products [36].

#### 3.9.2. Microstructure Analysis

Figure 8 shows the sample microstructure of different digestion stages. Initial emulsion (undigested), the microstructure was complete, and particle size was large. After the oral stage (pH 7.0), the particle size became small because of the dilution during measurement. After gastric digestion (pH 3.0), the samples showed a large aggregation, and the microsphere structure was destroyed, which may be caused by the enzymatic hydrolysis of proteins in the W phase. However, NaCas and XG have electrostatic repulsion because they have the same charge, which is sufficient in contact with pepsin and hydrolyzed in the gastric stage. After intestinal digestion (pH 7.0), a small number of aggregates still existed in the sample. It has been reported that the release of nutrients can be regulated by particle structure [37].

#### 3.9.3. SDS-PAGE Analysis

SDS-PAGE determined the molecular weights of W phase proteins at different digestion processes. The degree of hydrolysis at different digestion stages was compared to explore the relationship between the release of Ca^2+^ and the hydrolyzed polypeptides in W phase digestion. The result was determined in Figure 9. The protein bands of initial emulsion (undigested) concentrated in 14.4–31 kDa, and the large molecular weight bands disappeared after gastric digestion, indicating that NaCas was hydrolyzed by pepsin during gastric digestion. Large molecular bands appeared in intestinal digestion, which may be caused by the combination of XG and NaCas hydrolysates. The SDS-PAGE results can be analyzed in conjunction with the previous microstructural results, indicating that the W phase’s hydrolysis destroyed the shell-core structure of the S/O/W emulsion, and CaCO_3_ was released to become Ca^2+^ under the action of gastric acid. Digestion and hydrolysis of the W phase destroyed S/O/W emulsion microspheres, which slowed down the release of Ca^2+^ during gastric digestion and effectively alleviated gastric irritation. It is speculated that the hydrolysis-specific peptide structure can combine with calcium ions to form a soluble complex, thus promoting calcium absorption in the small intestine. Studies have found that amino acid mineral chelates can directly pass through the edge of villi brush in the small intestine to reduce biochemical processes in the body, with fast transport speed and good absorption, and can reduce the antagonism with mineral elements in the body [38,39].

#### 3.9.4. Determination of Free Calcium Release

During gastric digestion, CaCO_3_ will be converted into Ca^2+^ under the action of gastric acid and released for absorption by the human body. The release process is easy to cause gastric irritation [40]. In contrast to free CaCO_3_, S/O/W emulsions have a slow-release effect due to the hydrolysis of the W phase, which was also evidenced by the microstructure of the different digestion stages (Figure 8). The release of Ca^2+^ in free CaCO_3_ and S/O/W emulsion during gastric digestion is shown in Figure 10. Both free CaCO_3_ and S/O/W emulsions delivered CaCO_3_ released faster in the first 20 min of gastric digestion, and then the release tends to be gentle, but the release rate of free CaCO_3_ was significantly faster than that of S/O/W emulsions. The results showed that the S/O/W emulsion delivery carrier has a controlled release effect on CaCO_3_ and could effectively alleviate the gastric stimulation caused by consuming a large amount of gastric acid and releasing Ca^2+^ during gastric digestion. Studies have shown that the konjac gum S/O/W lipid microspheres gradually increase the viscosity during stomach digestion, allowing for the slow release of konjac gum in the gut to achieve satiety and thus reduce appetite [13]. Based on S/O/W technology to protect NRRL B-30514, it was found that the formation of powdered probiotics and beet pectin into microspheres could effectively improve the environmental tolerance and digestive characteristics of probiotics [11].

Analyzed the Ca^2+^ release rate of free CaCO_3_ and S/O/W emulsion delivery during gastric digestion, and it was not the linear model, including the release rate rising phase and stationary phase. Logarithmic transformation was used to transform the nonlinear model into a linear model in the rising phase of release rate, and constant values were used to fit the data in the steady phase of release rate. The equations were as follows:(1)$$f_{1}(t)\left\{\begin{array}{c} -0.1631-0.3674t\;\;\;\;\;\;\;\;\;\;0\leq t<1\\ -0.1631+0.3674t+29.3485lnt\;\;\;\;\;\;1\leq t<20\\ 90.4261\;\;\;\;\;\;\;\;\;\;\;\;\;\;\;\;20\leq t \end{array}\right.$$
(2)$$f_{2}(t)\left\{\begin{array}{c} -1.3208+0.1807t\;\;\;\;\;\;\;\;\;\;0\leq t<1\\ -1.3208+0.1808t+14.5217lnt\;\;\;\;\;\;1\leq t<100\\ 81.9044\;\;\;\;\;\;\;\;\;\;\;\;\;\;\;\;100\leq t \end{array}\right.$$

Equation (1) was the release model of free CaCO_3_, Equation (2) was the release model of NaCas-XG stabilized emulsions embedded with CaCO_3_. The Ca^2+^ release kinetics equation reflects the Ca^2+^ release rate during gastric digestion and provides a mathematical model for later predicting the Ca^2+^ release rate.

## 4. Conclusions

In this research, S/O/W emulsions delivered to CaCO_3_ were prepared by the binary composite of NaCas and XG. The concentration of XG and the system pH were optimized, and it was found that when the concentration of XG was 0.5 wt% and pH was 7, the particle size was smaller, the distribution was uniform, and the physical stability was improved. CLSM results showed that CaCO_3_ was embedded in the emulsions, and the particle size distribution was more uniform with increased XG concentration. Cyro-SEM microstructure showed that the addition of XG made the S/O/W emulsion form a filament-like connector with a relatively more stereoscopic structure. XRD results showed that the characteristic peak of CaCO_3_ was partially covered due to physical embedding, indicating that S/O/W emulsions have a specific embedding effect on CaCO_3_. Infrared and Raman analysis and molecular docking results confirmed electrostatic and hydrophobic interaction between sodium caseinate and xanthan gum, which was beneficial to the construction of the system and the improvement of stability. In the GIT digestion model, S/O/W emulsions released Ca^2+^ slowly in gastric digestion stage, which proved the targeted slow-release effect of the S/O/W emulsion delivery vector. The results showed that the S/O/W emulsions delivery system is an effective way to promote the application of CaCO_3_.

## Figures and Tables

**Figure 1 foods-11-02854-f001:**
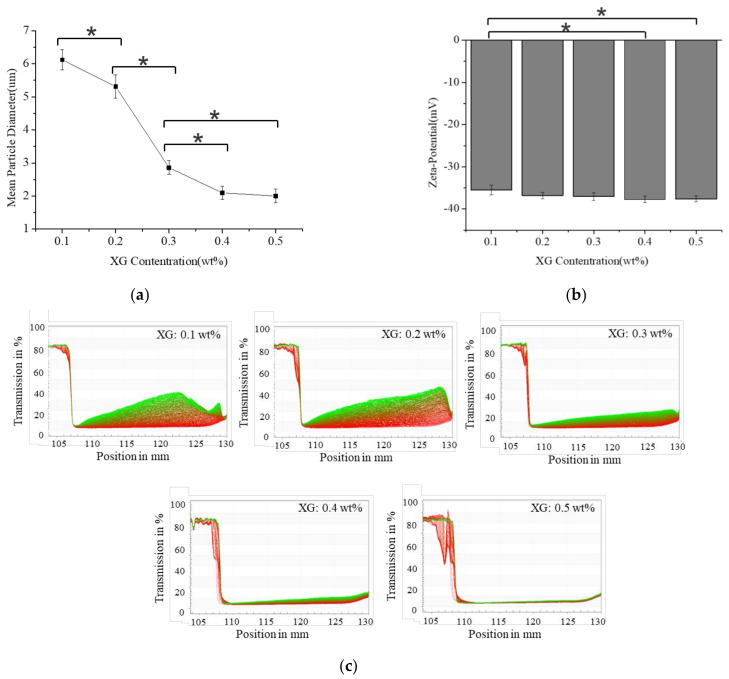
Effect of the different XG concentrations (0.1 wt%, 0.2 wt%, 0.3 wt%, 0.4 wt%, and 0.5 wt%) on the mean particle diameter, “*” indicates significant difference (**a**), zeta−potential, “*” indicates significant difference (**b**), and instability index (**c**) of CaCO_3_ S/O/W emulsions.

**Figure 2 foods-11-02854-f002:**
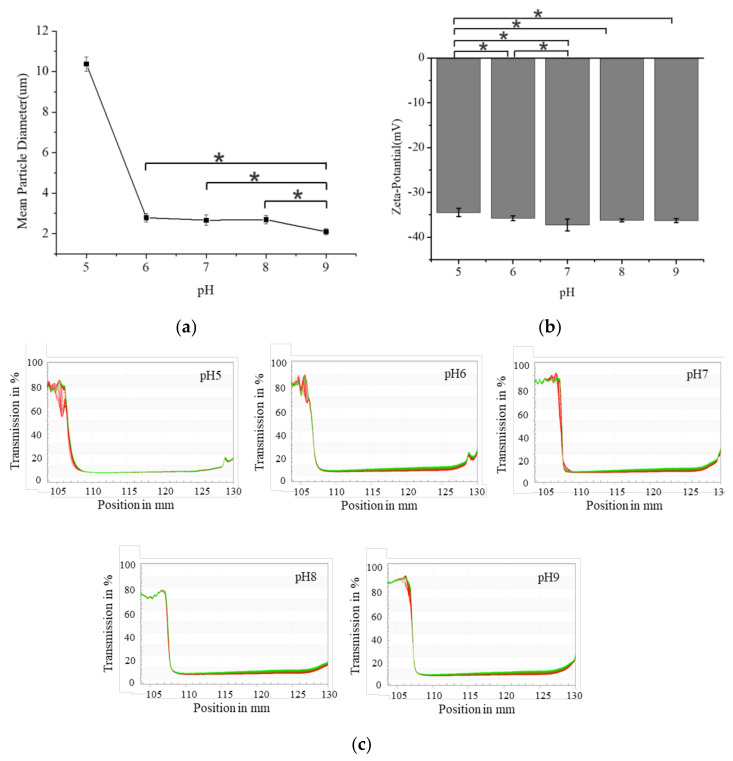
Effect of the different pH (5−9) on the mean particle diameter, “*” indicates significant difference (**a**), zeta−potential, “*” indicates significant difference (**b**), and instability index (**c**) of CaCO_3_ S/O/W emulsions.

**Figure 3 foods-11-02854-f003:**
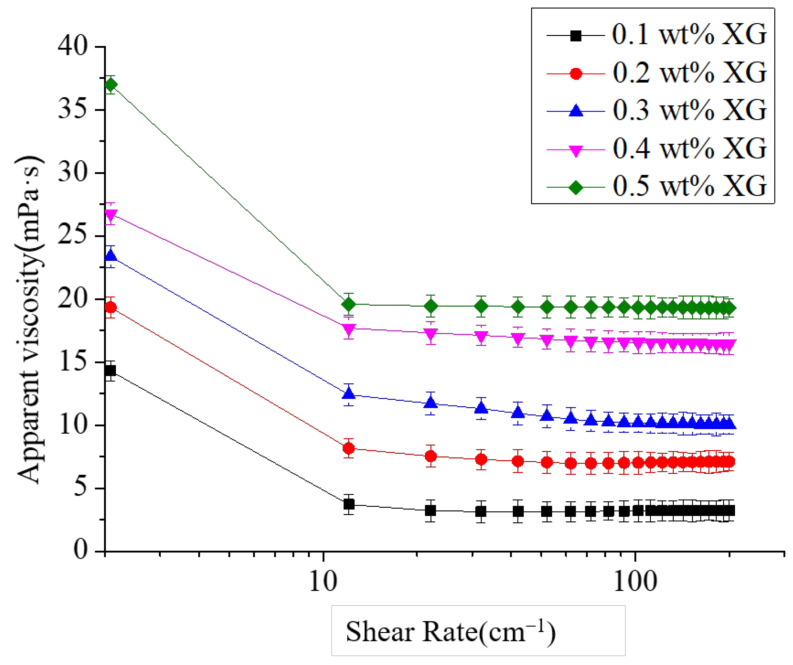
Effect of the different XG concentrations (0.1 wt%, 0.2 wt%, 0.3 wt%, 0.4 wt%, and 0.5 wt%) on the apparent viscosity of CaCO_3_ S/O/W emulsions.

**Figure 4 foods-11-02854-f004:**
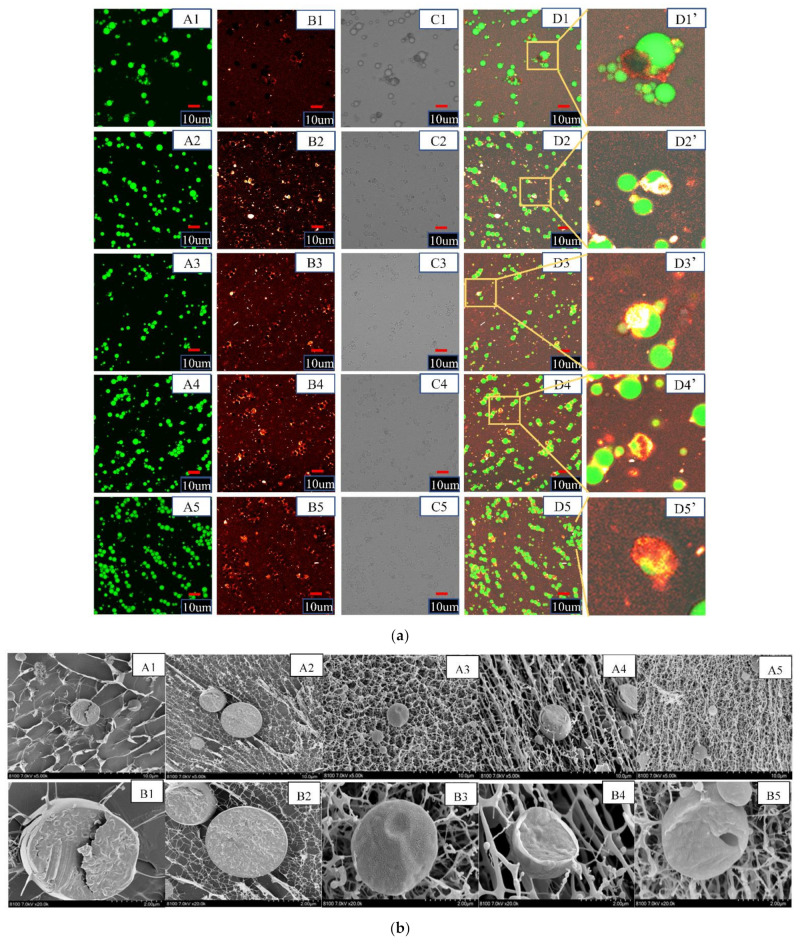
(**a**) Confocal laser scanning microscope images of the CaCO_3_ S/O/W emulsions stabilized by NaCas and different concentrations of XG mixtures. (A, Oil phase was stained with Nile red, excitation at 488 nm; B, Protein phase was stained with Nile blue, excitation at 635 nm (ii); C, bright field view; D, combined image; D’, high magnified image). Scale bar: 10 um. (1–5: 0.1, 0.2, 0.3, 0.4 and 0.5 wt% XG). (**b**) Cyro-SEM images of the CaCO_3_ S/O/W emulsions stabilized by NaCas and different concentrations of XG mixtures. (A, 5000×; B, 20,000×). (1–5: 0.1, 0.2, 0.3, 0.4 and 0.5 wt% XG).

**Figure 5 foods-11-02854-f005:**
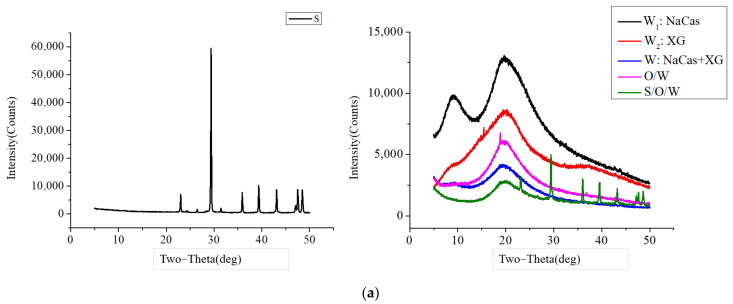
The XRD (**a**), Infrared spectral (**b**), and Raman spectrum analysis (**c**) of the CaCO_3_ S/O/W emulsions.

**Figure 6 foods-11-02854-f006:**
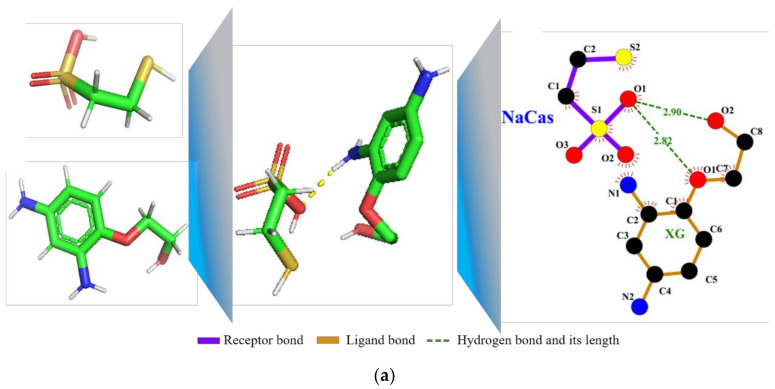
Molecular docking technique of NaCas and XG. (**a**) receptor-NaCas, ligand-XG; (**b**) receptor-XG, ligand-NaCas.

**Figure 7 foods-11-02854-f007:**
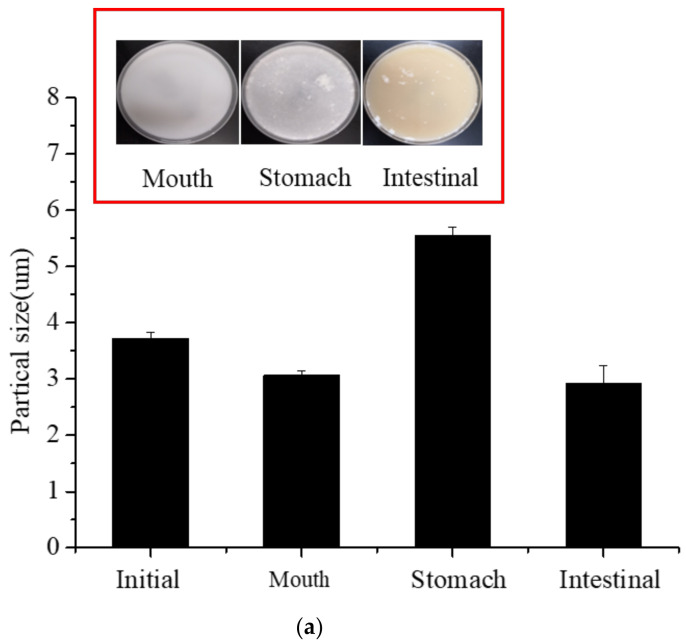
The particle size (**a**) and Zeta−potential (**b**) of the CaCO_3_ S/O/W emulsions at different stages of digestion.

**Figure 8 foods-11-02854-f008:**
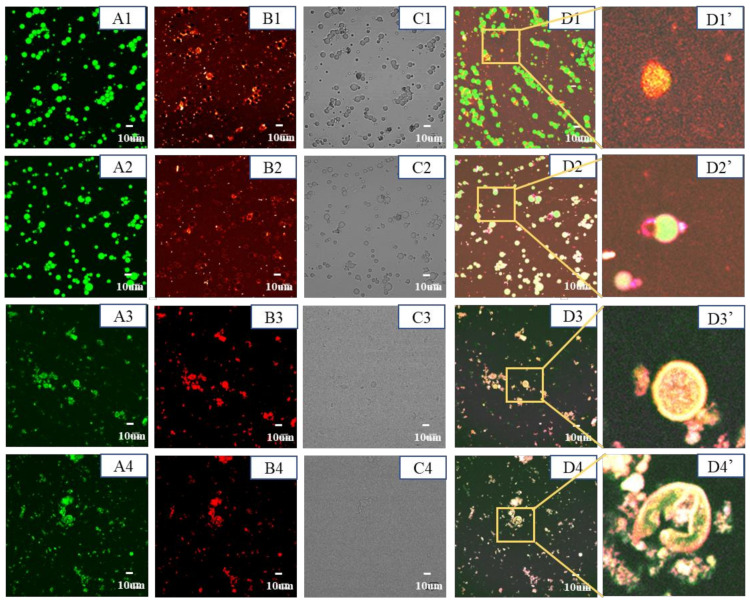
Microstructural analysis of in vitro simulated digestion of the CaCO_3_ S/O/W emulsions. (A, Oil phase was stained with Nile red, excitation at 488 nm; B, Protein phase was stained with Nile blue, excitation at 635 nm (ii); C, bright field view; D, combined image; D’, high magnified image). 1–4 represent different digestion stages of CaCO_3_ S/O/W emulsion: 1-initial emulsion; 2-oral digestive stage; 3-gastric digestive stage, 4-intestinal digestive stage.

**Figure 9 foods-11-02854-f009:**
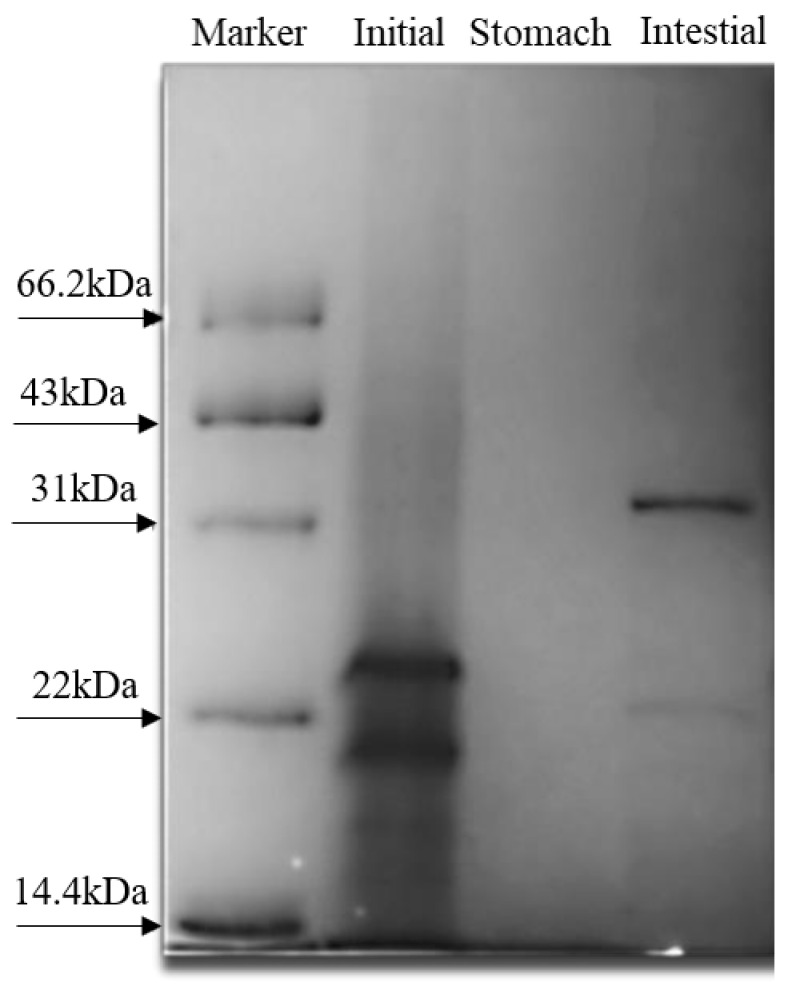
SDS-PAGE of the CaCO_3_ S/O/W emulsions at different stages of digestion.

**Figure 10 foods-11-02854-f010:**
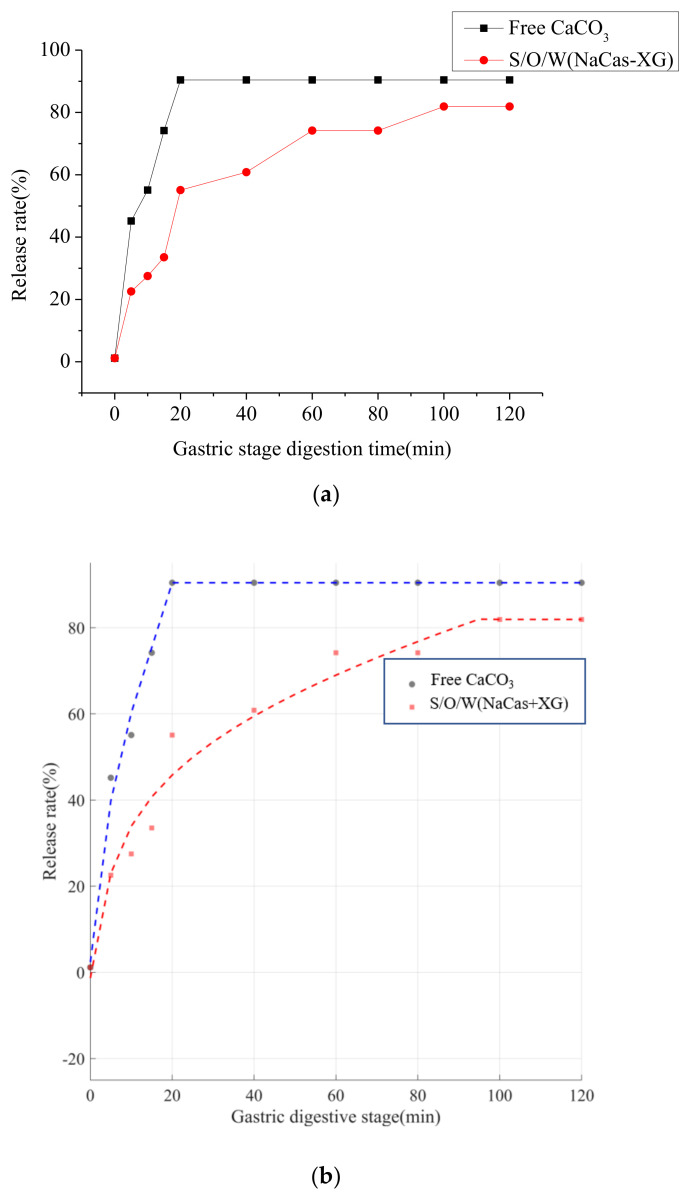
Ca^2+^ release rate curve (**a**) and fitting curve (**b**) of CaCO_3_ S/O/W emulsions in the gastric stage.

## Data Availability

The data presented in this study are available on request from the corresponding author.

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
