# Peer review of "Enhancing the Dispersion Stability and Sustained Release of S/O/W Emulsions by Encapsulation of CaCO3 Droplets in Sodium Caseinate/Xanthan Gum Microparticles"

_foods, 2022, doi:10.3390/foods11182854_

Round 1

Reviewer 1 Report

Dear authors

The research you present is significant and interesting. However, there is room to improve on the quality of the results and presentation.

Some key points that must be improved:

- you should check for statistical significance of the results, it is not sufficient to give means and standard deviations to say something is significantly different or not different. 

- the figure together with the figure caption should be self-explanatory, i.e. the reader should be able to understand the results by only studying hte figure and reading the figure caption (but not reading the manuscript text)

- there are sentences that miss verbs or are not understandable, e.g. line 54/55, line 65/66, line 149/150, line 156, 162

Details: 

Line 25: GIT has to be introduced first before you use the abbreviation

Line 25ff: it is unclear from this text that 0.5% XG is the highest addition. 

Keywords: abbreviations can not be used as keywords

line 106: put the word pepsin before the brackets

Line 115: soybean oil

Line 123: the zeta-potential of what was measured here, it is unclear from the tex

LIne 119: in which order did you add the two water phases?

Lines 134 / 173, 179: this wording does not work for a publication

LIne 194: Zeta-potential of the digested samples

Line 195: The experimental method was the same as for the undigested sample (see section 2.3)

Line 196/97: same issue

Line 209: what about statistical analysis to determine whether your results are significantly different from each other? Depending on the setup, the 3 repetitions might however not be sufficient to assess statistical significance

LIne 217: you can not say statistically significant without the proper statistical analysis

LInes 236/7: which results are the basis for this interpretation? 

Line 240: you can not say whether the results are significantly different from the given results

LInes 244 ff: either support the hypotheses with results or literature, please

Line 258: the sentence does not make sense as it is written here as there can not be a thickening within the particles, did you mean the emulsion?

Line 265: from the graph it can not be said whether you have shear thinning behaviour, it could also be a yield stress fluid. Working with a logarithmic scale might help

Line 300 onwards: it is very difficult to follow the text as the names of the phases are not the same in the text and the images

Section 3.6, 3.7: discussion is missing

Section 3.9.1: references??

Section 3.9.4: discussion and references missing completel

Captions (lines 549 onwards): you need to give clearer information here. E.g. Fig. 4: mention what 4a and what 4b is. Then say which image A1 - 5, E1 - 5 is what recipe. 

LIne 582: standard deviations not visible

Fig. 4a and 8: barely discernible as the images are very small

Figures overall: please find a way to aggregate certain figures into one, such as e.g. a and b etc. You currently have 24 figures which is simply too much. 

I hope that my inputs are helpful. 

Kind regards. 

Reviewer 2 Report

Dear authors, the manuscript "Enhancing the dispersion stability and sustained release of S/O/W emulsions by encapsulation of CaCO3 droplets in sodium caseinate/xanthan gum microparticles" is interesting, I leave some comments that could contribute to its improvement

Line 69: What do you mean by "continuous"?

Line 111: did you design the methodology? Were you based on any already published?

In general terms, you do not show a reference to where you based your methodologies on.

Line 209: The authors do not mention which statistical design they used

Line 210: At least in triplicate? It would be convenient for you to point out why some in triplicate and others in greater quantity

In general, their discussions are limited and lack references, Line 264, 278, 287, 399, among others

Line 426: correctly name the equations (equation 1 and 2), you could also reinforce the discussions

Round 2

Reviewer 1 Report

Dear Authours

Thank you for the careful build-in of my suggestions, the manuscript is in my eyes significantly improved. 

It would be great to add somewhere in the manuscript that the asterixs in the images show which results are significantly different from each other. 

Kind regards

Reviewer 2 Report

The authors have improved the work